# Mechanically Proving Guarantees of Generalized Heuristics:
# First Results and Ongoing Work

**Mohammad Abdulaziz,**[1,2] **Florian Pommerening,**[3] **Augusto B. Corrêa,**[3]

[1]King's College London, United Kingdom,
[2]Technische Universität München, Germany,
[3]University of Basel, Switzerland,
mohammad.abdulaziz@in.tum.de, {florian.pommerening,augusto.blaascorrea}@unibas.ch

## Abstract

The goal of generalized planning is to find a solution that works for all tasks of a specific planning domain. Ideally, this solution is also efficient (i.e., polynomial) in all tasks. One possible approach is to learn such a solution from training examples and then prove that this generalizes for any given task. However, such proofs are usually pen-and-paper proofs written by a human. In our paper, we aim at automating these proofs so we can use a theorem prover to show that a solution generalizes for any task. Furthermore, we want to prove that this generalization works while still preserving efficiency. Our focus is on generalized potential heuristics encoding tiered measures of progress, which can be proven to lead to a find in a polynomial number of steps in all tasks of a domain. We show our ongoing work in this direction using the interactive theorem prover Isabelle/HOL. We illustrate the key aspects of our implementation using the Miconic domain and then discuss possible obstacles and challenges to fully automating this pipeline.

## Introduction

*Generalized planning* is the problem of finding a solution that solves all tasks in a planning domain. Due to the PSPACE-completeness of classical STRIPS planning the performance of such solutions is usually unpredictable. However, for some domains polynomial solvers exist and can be synthesized automatically (e.g., Francès et al. 2019a). This is particularly interesting for situations where a given domain can be analyzed for some time before tasks of that domain have to be solved but time for solving the tasks is limited. For example, in space applications, domains can be analyzed on the ground before a mission, while calling a planner with potentially exponential running time during a mission is not feasible. Most methods to synthesize polynomial solvers are based on a supervised learning approach (Yoon, Fern, and Givan 2006; Arfaee, Zilles, and Holte 2011; Francès et al. 2019a; Shen, Trevizan, and Thiébaux 2020), where the solver is learned from example tasks in the domain. The performance guarantee then relies on the fact that the learned solver generalizes to the whole domain. So far, the proofs for this generalization have been manual pen-and-paper proofs. This has two problems. Firstly, the man-

ual proof can be cumbersome and detailed[1], especially given that it might depend on all the action schemata in the given domain. Secondly, due to the cumbersomeness of these manual proofs, they are error-prone, which might not suffice for resource-restrained applications, especially when such applications are safety-critical.

In this paper, we work towards a system where such proofs are machine-checked, as well as largely automated. Towards that end, we employ an interactive theorem prover. Interactive theorem provers involve a human in the process of mechanically proving a statement. The human provides high-level proof steps and the theorem prover tries to fill in the missing steps automatically. This ability to automate cumbersome proof steps and to check the correctness of the whole proof made interactive provers an obvious choice for many applications, where proofs are cumbersome and error-prone, and especially when the underlying proof obligation is undecidable. A notable example is program verification, where interactive theorem provers are the most successful method to verify large scale pieces of software, e.g. an operating system kernel (Klein et al. 2009), a verified compiler for C (Leroy 2009), and an LTL model checker (Esparza et al. 2013).

Here we present our work-in-progress on a framework to mechanize proofs of performance guarantees for generalized planning heuristics. In particular, we focus on *generalized potential heuristics* (Francès et al. 2019a), which are weighted sums of state features. In contrast to potential heuristics in classical planning (Pommerening et al. 2014), features are defined with a description logic and can be evaluated in all tasks of a domain. Francès et al. prove that several learned generalized potential heuristics (for different domains) lead to a backtrack-free search in any possible task of the domain where the heuristic value decreases by at least 1 in every step. Together with a guarantee on the maximal and minimal heuristic value, this ensures a polynomial search effort. However, all these proofs were done manually by the authors. This is clearly not an ideal scenario, as the ultimate goal of generalized planning is to come up with such heuristics and their performance guarantees without input or help from a user.

---

[1]The manual proofs by Francès et al. 2019b do not look that complicated because they leave out a lot of low-level details.

The framework we work on uses the interactive theorem prover Isabelle/HOL (Nipkow, Wenzel, and Paulson 2002) to formally verify these manual proofs. We develop a framework for interactive reasoning about annotated PDDL domains. Such annotations include *(i)* domain assumptions, i.e. properties that should hold for all instances of the domain, *(ii)* domain invariants, which are properties that follow from the action descriptions and domain assumptions for every reachable state in every instance of the domain, and *(iii)* generalized heuristics, which we prove to be descending. Given such an annotated domain, this framework is capable of parsing the domain, automatically proving many properties about it, and leaving only the core proof tasks for the user to be proven interactively. Not all parts of the framework that could be automated are automated yet and it is currently limited to generalized potential heuristics that form *tiered measures of progress* (Parmar 2002). As a proof of concept, we interactively completed the relevant proofs for the domain Miconic and were able to prove that the generalized heuristic computed for it using the method of Francès et al. indeed has the reported performance guarantee.

Together with a method to create hypothesis heuristics such as the one by Francès et al., our framework could be developed into an integrated tool to find solutions to a generalized planning task that come with a formally verified performance guarantee.

## Running Example: Miconic

Throughout the paper, we use the Miconic domain (Koehler and Schuster 2000) as a running example. In this domain, passengers start at different floors of a building and each passenger has one specific destination floor. The goal is to find a plan where an elevator transports each passenger to its destination.

Miconic can be seen as a logistics problem where a vehicle (the elevator) needs to transport items (passengers) to their desired destination. Francès et al. (2019a) showed that it is possible to synthesize a compact planner for this domain.

## Generalized Planning

We consider STRIPS planning tasks with negation represented in PDDL (McDermott et al. 1998; Haslum et al. 2019). A *PDDL planning task* is encoded using a first-order vocabulary and can compactly represent a state space.

The first-order vocabulary of a task defines a *domain*. A domain is a tuple $\Sigma = \langle \mathcal{P}, \mathcal{C}, \mathcal{A}, \mathcal{X} \rangle$, where $\mathcal{P}$ is a set of *predicate symbols*, $\mathcal{C}$ is a set of *constants* (i.e., nullary functions), $\mathcal{A}$ is a set of PDDL *action schemas*, and $\mathcal{X}$ is a set of first-order logic formulas, called *domain assumptions*, which we will discuss later. A lifted atom is a first-order atom $P(x_1, \ldots, x_n)$ where $P \in \mathcal{P}$ and all $x_i$ are either variables or objects from $\mathcal{C}$. A lifted literal is either a lifted atom or its negation. A PDDL action schema $a \in \mathcal{A}$ is represented with a precondition $pre(a)$ and a set of add effects $add(a)$, and a set of delete effects $del(a)$, where $pre(a)$ is a set of lifted literals and both $add(a)$ and $del(a)$ are sets of lifted atoms.

A *planning task* $\Pi$ in a domain $\Sigma$ is a tuple $\langle \mathcal{O}, \mathcal{I}, \mathcal{G} \rangle$, where $\mathcal{O}$ is a set of *objects*, $\mathcal{I}$ is the *initial state*, and $\mathcal{G}$ is the *goal condition*. A predicate symbol $P \in \mathcal{P}$ applied to constants in $\mathcal{C} \cup \mathcal{O}$ is a (ground) atom and a subset of atoms is a *state* with the interpretation that all atoms in the state are true, while all others are false. The initial state $\mathcal{I}$ is a state, while $\mathcal{G}$ is a set of atoms with the interpretation that our aim is to make all atoms in $\mathcal{G}$ true. Any state $s$ where $\mathcal{G} \subseteq s$ is a *goal state*. The set of domain assumptions $\mathcal{X}$ contains first-order formulas over $\mathcal{I}$ and $\mathcal{G}$ that restrict which instances we consider part of the domain.

Action schemas $a \in \mathcal{A}$ can be grounded by consistently replacing the variables in $pre(a)$, $add(a)$, and $del(a)$ with constants from $\mathcal{C} \cup \mathcal{O}$. If every atom in the precondition of a ground action is true in a state, we say the action is *applicable*. If an action $a$ is applicable in a state $s$ then applying it leads to the successor state $s[a] = (s \setminus del(a)) \cup add(a)$. The set of successor states of a state $s$ is $succ(s) = \{s[a] \mid a \text{ is applicable in } s\}$.

**Example 1** *In the Miconic domain, the set $\mathcal{P}$ has the following predicates: origin$(p, f)$, indicating that passenger $p$ starts at floor $f$; destin$(p, f)$, indicating that passenger $p$'s destination is floor $f$; boarded$(p)$, indicating that passenger $p$ has boarded into the elevator; served$(p)$, indicating that passenger $p$ has reached their destination; lift-at$(f)$ indicating that the elevator is in floor $f$; floor$(f)$ and passenger$(p)$ indicating the types of $f$ and $p$; and above$(f_1, f_2)$, indicating the order of floors.*

*The set $\mathcal{C}$ is empty for this domain and all constants are defined by the set of objects $\mathcal{O}$ of each instance.*

*There are four action schemas in the domain:* move *the elevator* up *or* down*; letting a passenger* board *at their origin or letting them* depart *at their destination. Moving the elevator from $f_1$ to $f_2$ requires and then deletes lift-at$(f_1)$ and adds lift-at$(f_2)$. A passenger can board an elevator on its origin floor which adds boarded$(p)$ (due to a modeling bug in the IPC domain there is this can be done even if passenger is already boarded). Similarly, a boarded passenger can depart only at their destination, which deletes boarded$(p)$ and adds served$(p)$. Additional conditions ensure that the action parameters have the correct types using the predicates floor$(f)$ and passenger$(p)$.*

*The domain assumptions $\mathcal{X}$ contain formulas restricting the instances we consider as part of the domain. For example, we use a domain assumption for the fact that the goal consists of serving passengers:*

$$\forall a \in \mathcal{G}. \exists p. (a = served(p) \land passenger(p) \in \mathcal{I}).$$

*Other examples would be the assumptions that initially the lift is at exactly one floor, and that every person has exactly one origin and one destination floor. Note that these assumptions are consistent with the original Miconic instances (Koehler and Schuster 2000).*

*Generalized planning* is the problem of finding solutions that work for any given task of a fixed domain (Jiménez, Segovia-Aguas, and Jonsson 2019). In other words, given a fixed domain $\Sigma$, we want to solve any possible task $\Pi$ of $\Sigma$.

# Generalized Potential Heuristics

A *heuristic* is a function $h$ mapping states to $\mathbb{R}_0$. A *generalized heuristic* is a function defined over all states of all possible tasks of a given domain. We use some terminology by Seipp et al. (2016) to characterize heuristics: a state is *alive* if it is not a goal state, reachable from the initial state, and a goal state is reachable from it. A heuristic $h$ is *descending* on a state $s$ if $s$ has at least one successor $s'$ where $h(s') \leq h(s) - 1$, and it is *dead-end avoiding* on a state $s$ if for every unsolvable state in $s' \in succ(s)$ $h(s) < h(s')$. A heuristic is descending and dead-end avoiding (DDA) if it is descending and dead-end avoiding on every alive state. Heuristics with this property guide standard greedy algorithms directly to a goal and are desirable in satisficing planning (Helmert et al. 2022).

Francès et al. (2019a) introduce the concept of *generalized potential heuristics*, which are weighted sums over features mapping states to integers. We limit both weight functions and features to natural numbers.

**Definition 1 (Generalized Potential Heuristics)** *Let $S$ be a set of states, $\mathcal{F}$ be a set of features $f : S \to \mathbb{N}$, and $w : \mathcal{F} \to \mathbb{N}$ be a weight function. The value of the generalized potential heuristic with features $\mathcal{F}$ and weights $w$ on a state $s \in S$ is*

$$h(s) = \sum_{f \in \mathcal{F}} w(f) \cdot f(s).$$

In our work, we are interested exclusively in *cardinality features* over *description logic* (DL) *concepts* and *roles*. We use the $\mathcal{SOI}$ language with equality role-value-maps (Baader et al. 2003). This language can be defined with the following production rules:

$$\begin{aligned}
C :=& \top \mid \bot \mid A_C \mid \{a_1, \ldots, a_n\} \mid \\
& (\neg C) \mid (C \sqcup C) \mid (C \sqcap C) \mid \\
& (\exists R.C) \mid (\forall R.C) \mid (R = R),
\end{aligned}$$

where $A_C$ is a set of *named concepts*, forming the basis of the inductive definition, and $a_1, \ldots, a_n$ are *nominals*. Similarly, roles are defined as follows

$$R := A_R \mid (R^{-1}) \mid (R \circ R) \mid (R^+),$$

where $A_R$ is a set of *named roles*. Due to space limitation, we point the reader to Francès et al. (2019a) and Baader et al. (2003) for detailed explanations on the semantics of each constructor above. Intuitively, in a state $s$ a named concept $A_C$ like *boarded* evaluates to the set of objects $o$, where $boarded(o) \in s$. The concepts $\top$ and $\bot$, $\{a_1, \ldots, a_n\}$ evaluate to the set of all, no, or the explicitly listed objects. The concepts $\neg C$, $C_1 \sqcup C_2$ and $C_1 \sqcap C_2$ have their intuitive interpretation using set complement, union and intersection. Roles evaluate to relations over objects, for example named roles $A_R$ like *origin* evaluate to the set of tuples $(p, f)$ where $origin(p, f) \in s$. The roles $R^{-1}$, $R_1 \circ R_2$ and $R^+$ evaluate to the inverse relation, relational composition and transitive closure of the recursive evaluations. The more complex concept $\exists R.C$ evaluates to the set of all objects $x$ such that there is an object $y$ such that $(x, y)$ is in the evaluation of $R$ and $y$

is in the evaluation of f$C$. The concept $\forall R.C$ is analogously defined and the concept $R_1 = R_2$ evaluates to all objects $x$ such that for all $y$, the tuple $(x, y)$ is either contained in both roles or in none of them.

The value of a feature $f = |c|$ is the cardinality of the concept $c$ evaluated in the respective state.

**Example 2** *The Miconic domain uses named concepts $A_C = \{served, boarded, lift\text{-}at\}$ and named roles $A_R = \{origin, destin\}$. We also consider the following concepts:*

$$\begin{aligned}
\textit{not-boarded-needs-lift} :=& \textit{passenger} \sqcap \neg \textit{served} \sqcap \neg \textit{boarded} \\
& \sqcap (\forall origin.\neg lift\text{-}at) \\
\textit{not-boarded-has-lift} :=& \textit{passenger} \sqcap \neg \textit{served} \sqcap \neg \textit{boarded} \\
& \sqcap (\exists origin.lift\text{-}at) \\
\textit{boarded-wrong-place} :=& \textit{passenger} \sqcap \neg \textit{served} \sqcap \textit{boarded} \\
& \sqcap (\forall destin.\neg lift\text{-}at) \\
\textit{boarded-right-place} :=& \textit{passenger} \sqcap \neg \textit{served} \sqcap \textit{boarded} \\
& \sqcap (\exists destin.lift\text{-}at) \\
\textit{passenger-served} :=& \textit{passenger} \sqcap \textit{served}.
\end{aligned}$$

*We can encode the following generalized potential heuristic $h$ using the concepts above:*

$$\begin{aligned}
h =& 5 \cdot |\textit{not-boarded-needs-lift}| \\
& + 4 \cdot |\textit{not-boarded-has-lift}| \\
& + 3 \cdot |\textit{boarded-wrong-place}| \\
& + 2 \cdot |\textit{boarded-right-place}| \\
& + 1 \cdot |\textit{passenger-served}|.
\end{aligned}$$

The generalized potential heuristic above encodes a *tiered measure of progress* (Parmar 2002). In words, the different concepts used in the heuristic describe different tiers, and an object of the task can only be part of one concept per state (i.e., the intersection of two concepts is empty). The idea is that these concepts are ordered – from best to worst – with respect to the goal condition. Moving one object from one concept to a better one *decreases* the heuristic value.

A heuristic encodes a tiered measure of progress if it is always possible to apply an action in a non-goal state to move an object from one concept to a better one. That is the case for the heuristic $h$ in Example 2. This implies that $h$ is descending for Miconic. Since every action moves at least one object to a better concept and the number of concepts is constant, a greedy search guided by $h$ will only take polynomial time in the number of objects (Francès et al. 2019a).

Currently, all proofs to show that a heuristic $h$ is a tiered measure of progress (and thus descending) are done manually (Parmar 2002; Yoon, Fern, and Givan 2006; Francès et al. 2019b). We work towards a system to automate this process.

## Approach

Tiered measures of progress guide a hill-climbing search directly to the goal, so they have guaranteed polynomial performance. However, so far, all methods that use measures of

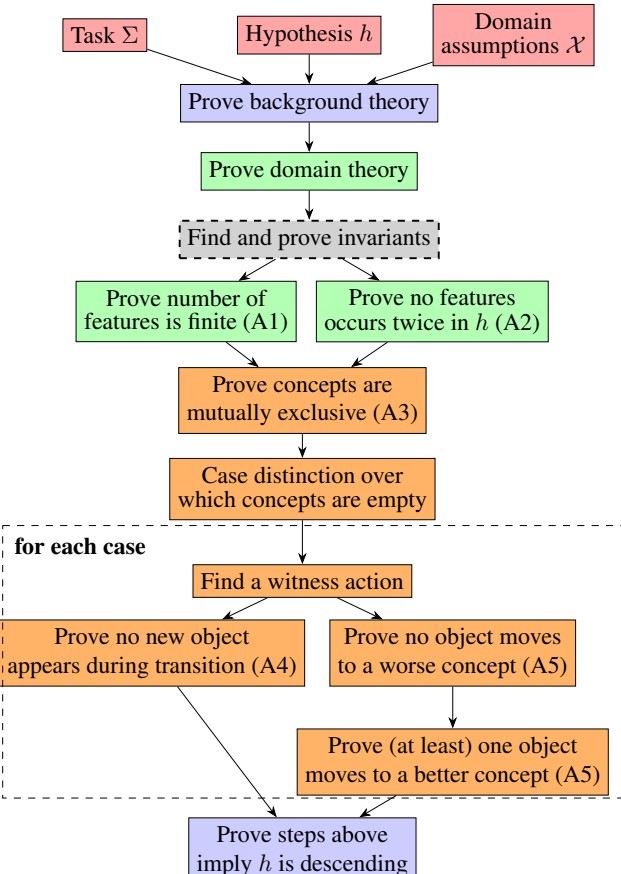

Figure 1: Steps of our proof. Red boxes represent the input given to Isabelle; blue boxes represent theories that need to be proved only once; green boxes represent parts that are fully automated; orange boxes represent proofs that are interactive. The part of the proof corresponding to the gray dashed box ("Find and Prove Invariants") is not currenctly implemented and it is assumed to be provided by the user.

progress either come up with them manually (Parmar 2002), or create them based on sample tasks (Yoon, Fern, and Givan 2006; Francès et al. 2019a). In both cases, it remains to show that the progress measure is a *strong measure of progress* for all tasks in the domain, i.e., that it generalizes beyond the sample tasks.

Our long-term goal is to have a system that can automatically generate a solver with a formally verified performance guarantees for a given domain. Such a guarantee on the performance is particularly important for systems that have to know in advance how much computational resources are required to solve a task, for example in space applications where battery power is severely limited. We show the first steps toward this goal. The system we envision would work in two phases:

1. Find a hypothesis, in our case a generalized potential heuristic that is descending on some sample tasks.

2. Prove (semi-)automatically that the hypothesis generalizes to all tasks in the domain, in our case, that the heuris-

tic is descending on all tasks in the domain. If this is not the case, generate more samples and start from phase 1 again.

Phase 1 is already handled by Francès et al. (2019a) whose method is guaranteed to find a heuristic that is descending on all sample tasks if such a heuristic exists. We work on Phase 2 here. In particular, we use automated and interactive theorem provers to show that a given generalized potential heuristic is indeed descending on all tasks of a domain. To do so, we make a case distinction and show in each case that there is a *witness action* – an action that is applicable in this case and only moves objects to *better* concepts. This in turn shows that the heuristic is a tiered measure of progress and thus descending in all tasks.

A fully automated process for this step is not possible, as some of the involved problems are undecidable in general. Interactive theorem proving involves the user in steps where no complete methods exist, or where a fully automated solution would require an impractical amount of time. Our hope is to automate as many steps as possible so the user can focus on the core problems. Since interactive proofs are checked by Isabelle/HOL (Nipkow, Wenzel, and Paulson 2002), the final result is a complete formally verified proof about the performance of the involved heuristic.

As we are reporting on work in progress, not all steps in the process are finished yet. In its current form, our implementation sets up a formal proof environment in the interactive theorem prover Isabelle/HOL. This implementation proves the statements we discuss in the rest of the paper and it is available online (Abdulaziz, Pommerening, and Corrêa 2022). Figure 1 gives an overview of the process. It also distinguishes between stepts that are interactive, fully automated, or not yet implemented.

As input, we take a PDDL description of the domain together with a description of domain assumptions and a list of invariants for the domain. PDDL files only describe the set of predicates $\mathcal{P}$, constants $\mathcal{C}$, and action schemas $\mathcal{A}$. Furthermore, we supplement it with the domain assumptions $\mathcal{X}$ and invariants. Similar to the domain assumptions, invariants are also first-order logic formulas. The idea is that if an invariant formula $I$ holds in a given state $s$ then it also holds in the successors of $s$. In contrast to domain assumptions, invariants need to be proved as consequences of the domain description and the domain assumptions. If an invariant holds for an initial state according to our assumptions $\mathcal{X}$, then it will hold on any reachable state (by induction).

**Example 3** *In Miconic, we use invariants to express that predicates like passenger are static, i.e., that if $p$ is a passenger in the initial state, $p$ will be a passenger in all reachable states. Another invariant expresses that served$(p) \in s$ implies boarded$(p) \notin s$ for all reachable states $s$. In total, we use ten such invariants in Miconic.*

For each invariant, we have an accompanying domain axiom claiming that the invariant holds in the initial state. The complete system would have to prove that each claimed invariant is actually invariant across the application of any action. Our prototype currently does not perform this step.

In addition to the domain and the invariants we also read

in a file containing a list of concepts and a description of the heuristic (the weights of involved concepts). We currently are limited to cardinality features but kept the system open to possibly extend this in the future. In the full system, the domain and domain axioms would have to be provided by the user, and the heuristic would be discovered in Phase 1, for example with the method by Francès et al. (2019a). Invariants could be discovered automatically but since there is an infinite number of invariants in each domain finding a sufficiently large set of invariants likely has to involve the user. We hope that in the future, we can set up the proof environment in such a way that failed interactive proof attempts fail with an error that helps the user identify missing invariants.

In the theory that we set up automatically, the user then has to prove the following properties interactively:

1. Given two concepts, show that they are mutually exclusive, i.e., there is no state where an object is in two of the concepts.

2. Prove that all invariants specified in the input files are indeed invariant, i.e., if they hold before an action application, they hold after the action application.

3. Possibly split cases further. We automatically set up a case distinction for alive states based on the concepts used in the heuristic. This automatic case distinction orders the concepts of the heuristic and has one case per concept where this concept is non-empty and every earlier concept is empty. The idea is that the concepts represent a tiered measure of progress and we distinguish cases based on the first non-empty tier in this progress measure. If another case split is required, this has to be done interactively.

4. In each case, specify a witness action that will lead to a decreasing heuristic value in alive states of this case or show a contradiction in the case. Specifying a witness action amounts to choosing an action schema and defining parameters of that action. The parameters have to be objects whose existence we can derive from the case. For example, if we handle a case where the concept $passenger \sqcap \neg served$ is non-empty, we can derive the existence of an object $p$ such that $passenger(p) \in s$ and $served(p) \notin s$. This object can then be used as a parameter in the witness action. Showing a contradiction in a case usually involves showing that all states in this case are goal states. For example, if the concept $passenger \sqcap \neg served$ is empty, we can use the domain axiom that the goal consists only of $served$ facts for passengers to show that all goals are satisfied. Since we make a case distinction over alive states, showing that all states in a case are goal states (and thus not alive) yields a contradiction.

Currently, we have interactive proofs for our Miconic use case for all of these steps except for proving the invariants. As the invariants we use are simple, we do not expect large problems proving them interactively, however, if we accept arbitrary first-order formulas as invariants, proving them is a semi-decidable problem, so an automated method will fail in some cases where the domain does not hold.

Our framework is limited to generalized potential heuristics that encode tiered measures of progress where the concepts are mutex. We view this as an interesting use case because an intuitive way of specifying a progress measure is to describe the different *stages* certain objects can be in (e.g., passengers first wait for a lift, then are about to enter a lift, wait inside the lift, are about to exit the lift, and finally arrive at their destination). The description of these stages expressed with a description logic naturally forms mutex concepts and a tiered measure of progress.

We discuss the individual steps of our framework in more detail in the following sections.

## Foundations for Formal Reasoning about Domains and Heuristics

One of our main contributions is constructing a formal mathematical background theory to aid in the formal reasoning about PDDL domains and formulas in the DL fragment we use. Such a background theory has to contain a formalization of the syntax and semantics of PDDL and the DL fragment in Isabelle/HOL's logic. The syntax of PDDL and the description logic is formalized in the form of an abstract syntax tree, and the semantics are formalized by defining functions and predicates (e.g. defining what is a valid plan, or what is the result of action execution) operating on the abstract syntax trees. Beyond the syntax and the semantics, this background theory contains general theorems which are reusable in any application that requires formal reasoning about PDDL and description logic.

### PDDL

For PDDL, we build on the work of Abdulaziz and Lammich (2018), who formalized its syntax and semantics in Isabelle/HOL. Their theory uses a more general definition of PDDL, though, and is not restricted to STRIPS tasks with negation. For example, in general PDDL action preconditions are first-order formulas, objects are typed, and effects can have additional conditions. In contrast, we focus on STRIPS, where preconditions are sets of literals, objects are untyped, and effects are given as two sets of atoms. The reasoning in the more general PDDL setting is not problematic in theory but can introduce some complications when formally proving theorems about a given STRIPS domain. When formalizing a proof in a STRIPS domain, reasoning about actions whose preconditions are sets of literals is more natural in interactive reasoning, as it more closely follows the pen-and-paper proofs. Furthermore, there is better support for automatically reasoning about sets in Isabelle/HOL. Likewise, reasoning about types is inconvenient when we work in STRIPS domains where objects are untyped.

To be able to reason at the proper level when formally reasoning about STRIPS problems we built a framework of formal theorems and automated proof methods. This framework transforms proof obligations about PDDL domains, which contain formulas as preconditions, into proof obligations about sets. It also can automatically discharge any proof obligations about object types when reasoning about STRIPS problem. In our experience, using this framework

significantly simplifies the process of formally proving statements about the Miconic domain.

In addition to changing perspective from general PDDL to STRIPS, our background theory defines the terms we introduced in the background section. In particular, we define what an alive state is and prove properties for alive states, for example that alive states are reachable and that invariants that hold in the initial state also hold in all alive states. In addition to being useful for reasoning about potential heuristics, this theory might find application in certifying unsolvability of planning tasks.

## Formalizing Concept Languages

Another contribution of our work is that we formalized the abstract syntax and the semantics of concept languages. For instance, Listing 1 shows the formalization of the abstract syntax of the fragment of description logic we consider. This listing shows how we use algebraic data-types, as implemented in Isabelle/HOL, to model the abstract syntax. Algebraic data types are widely used in functional programming[2]. They are used to elegantly recursively represent finite structures, e.g. trees. In our case, a role could be a goal predicate, an inverse of another role, a transitive closure of another role, or a composition of two roles. The latter three possibilities are recursively defined, i.e. they are defined in terms of roles. Also note that, next to each of the possibilities of the roles, we define a custom syntax which strongly matches the pen-and-paper format. Similar to roles, the abstract syntax tree of a concept is defined recursively. Isabelle/HOL can automatically compute a lattice for such types, thus showing that the data type is well-defined. Based on that lattice, it derives induction principles which can be used for proving theorems by structural induction on a variable with the given data type. The implementation in Isabelle/HOL for deriving these proofs is based on bounded natural functors (Traytel, Popescu, and Blanchette 2012).

The semantics, i.e. functions that assign meaning to the syntax are also formalized in Isabelle/HOL. The semantics of concepts are formalized in the form of a function that maps a state and a concept to a set of objects. For roles an analogous function maps to sets of pairs of objects. Listing 2 shows a sample of these functions. These functions are primitive recursive, i.e. they are defined recursively in terms of recursive algebraic data types. In particular, they are defined using *pattern matching*. For instance, the function role_value is defined to take, as arguments a role and a state. It is defined recursively in the role via pattern matching, where its value depends on the different possibilities a role could be. In all cases, this function returns a set of pairs of objects, defined using set comprehension. Note that the syntax used to define these sets is analogous to the pen-and-paper syntax one would use to define the sets. Similarly, we have another recursive function defining the value of a concept. Isabelle/HOL is able to automatically prove termination for all these recursive functions. Interested readers should consult Isabelle's documentation,

which summarises Isabelle/HOL's function definition facilities, and Krauss (2009) for the theory behind it.

A main focus of our formalization of description logic was to develop a library which enables better (semi)automated reasoning. Since automation methods in Isabelle/HOL are chiefly based on Gentzen-style deduction,[3] we prove lemmas about concept languages which can act as Gentzen-style deduction rules. For instance, the lemma in Listing 3 is the analogue of conjunction introduction, but for concept languages. In particular, that theorem is an implication, which is written as a right-arrow. The assumptions of that theorem, to the left of the arrow, lie between the square brackets and are separated by a semi-column. The conclusion of the theorem is to the right of the arrow. Again, we use the set element relation ($\in$) and the other functions were defined in the previous two listings.

In addition to these general lemmas aimed to make automation easier, we also formalized the definitions related to generalized potential heuristics. This includes the definition of the value of a feature and the heuristic in a given state, what it means to be decreasing transition, and what it means to be a descending heuristic. The overarching goal for this library is to provide background theory to aid in proving that a given heuristic is decreasing in a given domain. Our main theorem in this section sets out sufficient conditions to show that a transition is a *decreasing transition*, w.r.t. a given heuristic. To discuss it, we first have to define some additional notation.

We say an object *moves* from concept $C$ to $C'$ along a transition from state $s$ to a state $s'$ if the object is in concept $C$ in state $s$ and in concept $C'$ in $s'$. For the interpretation of a generalized potential heuristic as a tiered measure of progress, we define $C_h$ as the concepts used in the features of a heuristic $h$. We consider a partial order on $C_h$ based on the weight of the respective cardinality features in the heuristic. We say concept $C$ is *better* than concept $C'$ if it occurs lower in this partial order, i.e. if it occurs with a lower weight in the heuristic.

**Theorem 1** *A transition from a state $s$ to a state $s'$ leads to a decrease in the value of a generalized potential heuristic $h$ using concepts $C_h$ if*

*A1 the number of features in $h$ is finite,*

*A2 no feature occurs twice in the heuristic,*

*A3 the concepts in $C_h$ are mutually exclusive, i.e. no object can be in two different concepts of $C_h$ in any given state,*

*A4 there are no objects that are in some concept $C' \in C_h$ in state $s'$ but in no concept $C \in C_h$ in state $s$ (i.e., no objects "spontaneously appear" in some concept during the state transition),*

*A5 no objects move to worse concepts, and at least one moves to a strictly better concept.*

---

[2]Interested readers should refer to standard textbooks, like the book by Thompson (2011, Chapter 15).

[3]These are proof systems in which propositions are proved using tree-like proofs. Every proof step is represented as a branching in the proof tree using one deduction rule. Interested readers can consult the book by Troelstra and Schwichtenberg (2000, Chapter 3), which provides a good introduction.

Listing 1: Abstract syntax of a concept language.

```
      datatype role =
2     PredRole predicate
      | GoalPredRole predicate
4     | Inverse role                  ("_c⁻¹" 29)
      | TransitiveClosure role        ("_c⁺" 29)
6     | Composition role role         (infix "oᶜ" 25)

8     datatype concept =
      Universe                        ("⊤c")
10    | Empty                         ("⊥c")
      | PredConcept predicate
12    | GoalPredConcept predicate
      | Negation concept              ("¬c _" [40] 40)
14    | Union concept concept         (infix "⊔c" 30)
      | Intersection concept concept  (infix "⊓c" 35)
16    | RestrictEx role concept       ("∃c_._" 10)
      | RestrictFa role concept       ("∀c_._" 10)
18    | RoleValueMap role role        (infix "=c" 50)
```

Note that we do not forbid objects from "spontaneously disappearing". This is not a problem as our weights are non-negative so objects leaving all concepts of $C_h$ can only decrease the heuristic value.

The proof of this theorem is moderately involved and takes around 500 lines in Isabelle. The core argument involves performing a structural induction on the heuristic itself, i.e. an induction on the set of weight/feature pairs that constitute the heuristic, generalizing over the heuristic function as well as its values in the states $s$ and $s'$.

The main goal of proving this theorem is to prove that a heuristic is descending for a given domain. The final theorem in our background theory states that if A1–A3 are satisfied and for every alive state in the domain there is a successor satisfying A4–A5, the heuristic is descending. This leaves A1–A5 as the main proof obligations for the user to show that a given heuristic is descending for a given domain.

### Domain- and Heuristic-Specific Reasoning

The theorems we described so far (Theorem 1 and all other theorems about PDDL, DL, and concept languages) are independent of a specific domain or heuristic. We now describe another important part of our contribution that supports formal reasoning about a given heuristic and domain. Our tools can automatically generate definitions for a PDDL domain and a heuristic as an Isabelle/HOL theory. These definitions are the representation about which the user reasons and proves theorems. Our tooling also automatically proves many properties about the domain and the heuristic. These properties are then used in the interactive proofs done by the user.

### PDDL

Given a domain, described in a PDDL file, we built tooling that parses the different elements of the domain into Isabelle/HOL definitions. In addition to the PDDL domain, we also parse domain invariants and domain assumptions. Each invariant is parsed into a domain assumption that the statement holds for the initial state and a theorem stating that this is indeed an invariant. In the current state of our systems, these statements about the invariants are not proved. However, the invariants we use for the Miconic example are very simple statements and we do not expect major difficulties proving them interactively. In general, proving invariants is semi-decidable, as any first-order logic formula can be written as an invariant. To express the domain assumptions, we make use of a feature in Isabelle/HOL called a *locale* (Ballarin 2014). Locales are a way to structure theories in Isabelle/HOL. In our context here, the most interesting aspect of locales is that assumptions specified in a locale are implicit assumptions for any theorem within the locale. We prove all our theorems within the domain's locale, so all domain assumptions are implicit assumptions to all theorems in the locale, i.e. to all the theorems about the given domain.

In addition to representing the domain, the invariants, and the domain assumptions, we also automatically prove technical lemmas that are needed to ease the automation of the proofs. For instance, we generate lemmas stating that all predicates in the domain have different names.

### Generalized Potential Heuristics

Similar to what we do with the PDDL domain, our system also parses the given description of the concepts and the heuristic and translates them into Isabelle/HOL definitions. We also prove many technical lemmas about the concepts in the given domain. In particular, we prove that concepts and features have different names. We also describe the different concepts in features of the heuristic, and the partial order between them based on the weights.

To prove that the given heuristic is descending, we show that the assumptions of Theorem 1 apply to the domain and the heuristic. Assumptions A1 and A2 are easy to show automatically for the given heuristic. Automating the proof of assumption A3 is more problematic. Showing that two concepts $C_1$ and $C_2$ cannot overlap is equivalent to showing that

Listing 2: Semantics of a concept language.

```
     fun role_value::"role ⇒ state ⇒ (object × object) set" where
2    "role_value (PredRole p) M =
        {(x,y) : untyped_object_tuples. (Atom (predAtm p [x,y])) ∈ M}" |
4    "role_value (GoalPredRole p) M =
        {(x,y) : untyped_object_tuples. (predAtm p [x,y]) ∈ (atoms (goal P))}" |
6    "role_value (Inverse r) M =
        {(x,y) : untyped_object_tuples. (y,x) ∈ (role_value r M)}" |
8    "role_value (TransitiveClosure r) M =
        {(x,y) : untyped_object_tuples. (x,y) ∈ (role_value r M)\<^sup>+}" |
10   "role_value (Composition r1 r2) M =
        {(x,z) : untyped_object_tuples.
12            ∃y. (x,y) ∈ (role_value r1 M) ∧ (y,z) ∈ (role_value r2 M)}"

14   fun concept_value::"concept ⇒ state ⇒ object set" where
     "concept_value Universe M = untyped_objects" |
16   "concept_value Empty M = {}" |
     "concept_value (PredConcept p) M = {x : untyped_objects. (Atom (predAtm p [x])) ∈ M}" |
18   "concept_value (GoalPredConcept p) M =
        {x : untyped_objects. (predAtm p [x]) ∈ (pos_atoms (goal P))}" |
20   "concept_value (Negation c) M = untyped_objects - (concept_value c M)" |
     "concept_value (Union c1 c2) M = (concept_value c1 M) ∪ (concept_value c2 M)" |
22   "concept_value (Intersection c1 c2) M = (concept_value c1 M) ∩ (concept_value c2 M)" |
     "concept_value (RestrictEx r c) M =
24      {x : untyped_objects. ∃y. (x,y) ∈ (role_value r M) ∧ y ∈ (concept_value c M)}" |
     "concept_value (RestrictFa r c) M =
26      {x : untyped_objects. ∀y. (x,y) ∈ (role_value r M) ⟶ y ∈ (concept_value c M)}" |
     "concept_value (RoleValueMap r1 r2) M =
28      {x : untyped_objects. ∀y. (x,y) ∈ (role_value r1 M) ⟷ (x,y) ∈ (role_value r2 M)}"
```

Listing 3: Conjunction introduction for our concept language.

```
     lemma concept_valueI:
2    "⟦x ∈ concept_value c1 M; x ∈ concept_value c2 M⟧ ⟹ x ∈ concept_value (c1 ⊓c c2) M"
```

the concept $C_1 \sqcap C_2$ is not satisfiable. This problem is undecidable for most fragments of description logic that include role-value maps (i.e., concepts like $(R_1 = R_2)$) (Schmidt-Schauß 1989). Without role-value maps, our fragment of description logic is contained in $\mathcal{ALCIO}_{reg}$ where satisfiability checks are EXPTIME-complete (De Giacomo 1995) and is thus decidable.

For domains where a descending generalized potential heuristic can be expressed without role-value maps (e.g. this is the case in Miconic), a complete decision procedure could be implemented in Isabelle/HOL to automatically prove that concepts are mutually exclusive. Currently, we do not use such a complete decision procedure but rely on an incomplete method of Isabelle/HOL. This incomplete method successfully discharges the proof of A3 for Miconic. However, we are not sure whether this would work for other domains. An additional complication could be that proving A3 needs the domain assumptions – as we stated, every theorem we prove has the domain assumptions as implicit assumptions due to our locale. This would be problematic as domain assumptions are arbitrary first-order formulas, so proving A3 will then be a first-order theorem proving task.

## The Interactive Part: Showing the Existence of an Improving Transition

The remaining part of the proof is to show that assumptions A4 and A5 of Theorem 1 hold for every alive state. Together with the proofs of A1–A3 described in the previous section, we can then conclude that in every alive state there is an action whose application decreases the heuristic value.

We prove assumptions A4 and A5 by case analysis. In particular, we use an ordering $C_1, C_2, \ldots, C_n$ over the concepts occurring in the heuristic description. We then have $n$ cases, where each case $1 \le i \le n$ assumes that concepts $C_1, C_2, \ldots, C_{i-1}$ are empty, and concept $C_i$ is not empty in a fixed alive state $s$. We prove automatically that at least an object exists in $C_i$ in state $s$. For this object, we automatically prove properties based on the concept that it is in.

**Example 4** *For concept "not-boarded-has-lift", we show that an object $p$ exists that satisfies*

$$passenger(p) \in s, \quad served(p) \notin s, \quad boarded(p) \notin s, \text{ and}$$
$$\exists f.(origin(p, f) \in s \land \textit{lift-at}(f) \in s).$$

Currently, we define these properties of the witness object

manually, but they closely follow the structure of the concept and could be derived by straightforward syntactic analysis.

The user then chooses an action and interactively proves that it is a witness action, i.e., that the successor reached through this action satisfies A4 and A5. Choosing an action consists of selecting an action schema and picking its parameters. The witness action in general cannot be derived automatically. While it would be possible to test all different action schemas, selecting the parameters is not as straightforward. For example, a parameter could simply be the object from the non-empty concept, but in general its existence could also be derived from nested existentially quantified concepts or domain assumptions.

Next, the user has to show that the precondition of the witness action is always satisfied in any alive state of the current case. In this proof, the user can use the domain assumptions, the assumptions about the current case (i.e., which concepts are empty/non-empty), and the properties of the object in the non-empty concept.

Finally, the user has to show three facts about the transition induced by applying the witness action in an alive state satisfying the current case:

1. for each concept in the heuristic, if an object was in this concept before applying the action, afterwards it is not in a worse concept.

2. at least one object moves to a better concept.

3. no object is added to any concept that was not in a concept before. To show that, we show that for each heuristic concept that, if an object is in the concept after the application, then it was in a heuristic concept before the action application.

Again the proofs can use the domain and case assumptions, properties of the object in the non-empty concept, but additionally they can use the fact that the precondition of the witness action is satisfied to derive further information about the state.

We automated the proof of each of the three proof obligations we described above. We implemented proof methods within Isabelle using the Eisbach framework (Matichuk, Murray, and Wenzel 2016) to automate these proofs. We have no theoretical guarantees on the completeness of these methods, but a single proof method works for all cases of Miconic.

Together, the three properties show assumptions A4 and A5 which allow the user to apply Theorem 1 to conclude that the heuristic value decreases while applying the witness action.

The last step in the proof, which the user has to perform interactively, is to combine all the previous results to prove that the heuristic has a decreasing successor in every alive state. If the default case split was used, this part of the proof would always look the same so it can be generated automatically. This then shows that the heuristic is a tiered measure of progress. This can in turn be used with the theorem we discussed earlier to show that the heuristic is indeed descending.

## Conclusion

We showed first results of our tool to automate proofs of performance guarantees using generalized heuristics in Isabelle/HOL. In particular, we show an interactive theorem proving method for planning domains and generalized potential heuristics. So far, our scope is limited to those heuristics representing tiered measures of progress expressed with mutually exclusive concepts. We showed the viability of the system by proving that a heuristic for the domain Miconic is indeed descending.

The next steps of our work are to automate some of the remaining parts of the proofs where possible. For example, proving that all concepts are mutually exclusive could be automated if we restrict the description logic to no longer allow role-value maps. Finding parameters for a witness action is an interesting problem as well. While it is potentially semi-decidable, incomplete methods that systematically try objects based on the non-empty concept and the domain assumptions could already cover many interesting cases.

In the long run, our framework could be used more generally for other proofs about PDDL domains and generalized heuristics.

## Acknowledgments

This work was supported by the Swiss National Science Foundation (SNSF) as part of the project "Certified Correctness and Guaranteed Performance for Domain-Independent Planning" (CCGP-Plan) and by the Deutsche Forschunggemeinschaft Reinhart Koselleck project NI 491/16-1 "Verifizierte Algorithmenanalyse". Moreover, this research was partially supported by TAILOR, a project funded by the EU Horizon 2020 research and innovation programme under grant agreement no. 952215.

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
