# OpenReview forum: "Mechanically Proving Guarantees of Generalized Heuristics: First Results and Ongoing Work"
_icaps-conference.org/ICAPS/2022/Workshop/HSDIP — HSDIP 2022_

### Official Review · Reviewer_GUAg · 2022-04-25
**Interesting work, worth discussion**

**Confidence:** 3
**Overall Score:** Accept

**Review:**


The paper presents a method of using a theorem prover to fill in the gaps in the proofs that a hypothesis generalized potential heuristic is indeed descending.
It is an interesting work, still work in progress, but could be an interesting addition to HSDIP program. In what follows, I present some comments/questions.

1. "So far, the proofs for this generalization have been manual pen-and-paper proofs": It would be good to have a citation with such an example, especially in light of the following "Secondly, due to the cumbersomeness of these manual proofs, they are error-prone".
2. In Miconic example, it would be great to mention what are the actions in the domain. That way, you would remove an ambiguity about what version of Miconic are you referring to.
3. Is unsolvable state the same as dead-end state? A state from which a goal cannot be reached? I know that this is not your terminology, but since you have decided to use it, why is it called dead-end avoiding if a dead end has higher h value than the parent? What if all other non-dead-end successors have even higher h values?
4. Example 2, the heuristic does not seem to be goal-aware. Why do you need to assign a non-zero weight to served passengers?
5. Theorem 1: Can objects "spontaneously disappear"? If so, please give an example. I presume that this relates to the union of concepts not fully covering all states. If not, is it not important for the theorem?
6. What is "The final theorem in our background theory"?
7. The proof of Theorem 1 is missing.
8. The generalized potential heuristic value as defined here (Definition 1) is not bounded from either above or below. While bounding from above might not be needed for descending heuristics, bounding from below is crucial. However, as far as I could see, you don't talk about it at all.
9. It is up to you, of course, but the word "Mechanically" in the title is often associated with something that was done so many times that is now done "without thought", automatically. I suppose that wasn't your intention with this title?


Minor:
1. Example 2: "lifted-at" -> "lift-at"
2. "The theorems" there is a single theorem in the entire current version.

---

> ### Author Response · Authors · 2022-04-29
> **Response**
>
> Thank you for your thoughtful review.
>
> Before answering the other questions, let us point out a mistake you spotted in the paper: As you mentioned in Q8, we missed a bound and we only consider feature weights that are natural numbers, not real numbers. This implicitly defines a lower bound of 0 for the heuristic value and influences our answers to your other questions. We'll fix Definition 1 in the paper.
>
> Q1) The pen-and-paper proofs we were referring to are in the following technical report (we'll add a reference to the text):
> https://ai.dmi.unibas.ch/papers/frances-et-al-tr2019.pdf
> The proofs are not that cumbersome there but that is because they are not spelled out in full detail and doing so would be difficult to do correctly.
>
> Q2) We'll add details about this.
>
> Q3) Yes, unsolvable states are the same as dead-end states. Dead-end avoidance only makes sense in conjunction with descending heuristics. In that case, not all non-dead-states (i.e., alive states) can have higher h values.
>
> Q4) The kind of heuristics we consider do not have to be goal-aware. In this particular case, we could make it goal-aware by weighting served passengers with 0. We chose this version of the heuristic because it more clearly shows a progression of objects (passengers) through the tiers of the progress measure (objects do not disappear from concepts in the heuristic at any time).
>
> Q5) Yes, objects can disappear, for example if we would not include the concept for served passengers (see Q4), a passenger leaving the elevator would disappear from concept "boarded-right-place" and not appear in a different concept. This is not a problem for the proof because of our restriction to non-negative weights. An object disappearing form the heuristic concepts can only decrease the heuristic value.
>
> Q6) The final theorem we refer to is exactly the one stating that if A1-A3 are satisfied and if A4-A5 hold for at least one successor of every alive state, then the heuristic is descending. The confusion is may be caused by the word "shows" in this sentence. We did not mean that there is a theorem that implies this fact, rather the theorem is the fact itself.
>
> Q7) The proof of Theorem 1 is too long to include in the paper (please see our response to the other review) and the formal proof of this statement within Isabelle is one of our contributions. We will make the full proof available as the Isabelle files but we are not sure what to include in the paper.
>
> Q8) Thank you for pointing this out. Our Isabelle files use natural numbers for weights giving an implicit bound. Definition 1 in the paper mistakenly uses real numbers instead because we based this definition on previous work.
>
> Q9) We understand the concern but "mechanized proof" is an established term in the area of theorem proving for the kind of proofs in our Isabelle theory.

---

### Official Review · Reviewer_dbfU · 2022-04-26
**Preliminary but great research direction**

**Confidence:** 3
**Overall Score:** Accept

**Review:**

This paper proposes an interactive mechanism to prove that the design of a given potential heuristic is dead-end avoiding and descending for any state in a given domain, which used along with Hill climbing, would render a domain polynomially solvable.

The paper presents preliminary ideas but it is a great research direction to explore further.

Given that I'm more familiar with planning concepts and less with description logic, which would be the case for the audience of the workshop, I would recommend the following suggestions, which may improve the clarity for this specific audience:

- Explain the semantics, at least informally, of the terms between definition 1 and example 2.

- Example 3: include the ten invariants used.

- The three Listings are not self-contained and need to be explained in the main text.

- Include a graph or flow chart to illustrate the steps of the proof, indicating which ones are automated and which ones require the user.

- Include the full synthesized proof following Theorem 1 for Miconic. This will finish the illustration of Miconic's proof. Otherwise, the proof for Miconic is not complete.

Minor comment:

- "Above" predicate should have 2 parameters relating the order of floors (Example 1).

---

> ### Author Response · Authors · 2022-04-29
> **Response**
>
> Thank you very much for your comments and suggestions. We will try to apply all the recommended changes.
>
> Regarding your last point about Theorem 1 for Miconic, we did not fully understand what you meant with the "full synthesized proof". The full proof would be the Isabelle theory file which we intend to publish with the paper but it consists of over 1000 lines and would be too large to include in the paper. In the Isabelle theory file, we have a general proof of Theorem 1 (independent of any domain) and additionally proofs that properties A1-A5 hold for Miconic specifically. Could you please explain in more detail what you were referring to?

---

> > ### Comment · Reviewer_dbfU · 2022-04-30
> > **Response - Miconic proof**
> >
> > Thanks for clarifying this point. I assumed the smallest instance of Miconic would require much less than 1000 lines from Isabelle. If that's not the case, including the link to the file would do.

---

> > > ### Author Response · Authors · 2022-05-02
> > > **Miconic proof**
> > >
> > > We'll include the link and make it clearer that the formal proof consist of those files.
> > >
> > > Since you mentioned "the smallest instance" of Miconic, we want to clarify another point: we do not have one proof per instance in Miconic. Instead, the Isabelle files have a single proof for the whole domain (independent of an instance). One of the advantages of using potential heuristics based on description logic features is that we can prove statements once that will be valid in all instances of the domain without proving them for each instance individually.